# Scratchpad Thinking: Alternation Between Storage and Computation in Latent Reasoning Models

**Sayam Goyal, Brad Peters, María E. Granda,**
**Akshath V. Narmadha, Dharunish Yugeswardeenoo, Cole Blondin,**
**Callum S. McDougall, Sean O'Brien, Ashwinee Panda, Kevin Zhu**

Algoverse AI Research

samnews07@gmail.com, bjpeters@mit.edu, me.granda25@gmail.com,
akshathvn27@gmail.com, dharyugi@gmail.com,
cole@algoverseairesearch.org, cal.s.mcdougall@gmail.com, seobrien@ucsd.edu,
ashwinee@princeton.edu, kevin@algoverseairesearch.org

## Abstract

Latent reasoning language models aim to improve reasoning efficiency by computing in continuous hidden space rather than explicit text, but the opacity of these internal processes poses major challenges for interpretability and trust. We present a mechanistic case study of CODI (Continuous Chain-of-Thought via Self-Distillation), a latent reasoning model that solves problems by chaining "latent thoughts." Using attention analysis, SAE based probing, activation patching, and causal interventions, we uncover a structured "scratchpad computation" cycle: even numbered steps serve as scratchpads for storing numerical information, while odd numbered steps perform the corresponding operations. Our experiments show that interventions on numerical features disrupt performance most strongly at scratchpad steps, while forcing early answers produces accuracy jumps after computation steps. Together, these results provide a mechanistic account of latent reasoning as an alternating algorithm, demonstrating that non linguistic thought in LLMs can follow systematic, interpretable patterns. By revealing structure in an otherwise opaque process, this work lays the groundwork for auditing latent reasoning models and integrating them more safely into critical applications. All code, data, and other artifacts will be publicly released upon acceptance.

## 1 Introduction

Latent reasoning models, such as CODI (Continuous Chain-of-Thought via Self-Distillation), offer a more efficient alternative to explicit Chain-of-Thought by performing multi-step inference in a continuous hidden space rather than generating text [Shen et al., 2025]. However, this efficiency comes at a steep price: transparency. By replacing an auditable textual trace with unobserved "latent thoughts," these models create a critical safety blind spot, making it difficult to verify their reasoning, align their behavior, or detect potentially dangerous emergent capabilities. Addressing this challenge is not just intellectually interesting; it is a prerequisite for safely deploying these powerful models in reliable applications.

To address this opacity, we turn to mechanistic interpretability, a field focused on reverse-engineering neural networks from their internal components [Bereska and Gavves, 2024]. This paper applies these techniques to produce a detailed mechanistic account of CODI's internal algorithm, using:

- **SAE-based analysis and steering** to isolate and identify how specific features are represented and used within CODI's latent space [Bricken et al., 2023].

39th Conference on Neural Information Processing Systems (NeurIPS 2025).

- **Activation Patching** to swap the model's residual stream at every layer for all the latent thoughts to identify components of the model responsible for specific tasks Meng et al. [2022], Heimersheim and Nanda [2024].
- **Early Answer Generation** to determine how early the model is able to arrive at the correct answer Deng et al., 2023).
- **Attention Analysis** to understand a model's decision-making process by examining which parts of the input sequence it focuses on when performing a specific computation [Rai et al., 2024].
- **Latent Step Manipulation** to causally probe the model's internal algorithm by actively altering its hidden reasoning steps to test their specific function, necessity, and sequential importance Meng et al. [2022], Heimersheim and Nanda [2024].

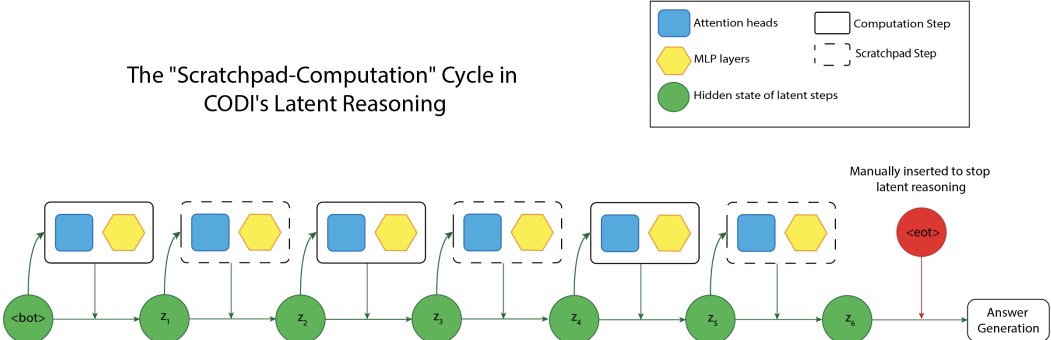

Figure 1: A simplified representation of CODI's reasoning process, illustrating our central finding. The model alternates between Computation Steps (solid outline), where operations are performed, and Scratchpad Steps (dashed outline), which store and represent key numerical information for subsequent access.

Our evidence reveals the model executes an alternating "scratchpad-computation" cycle as seen in Figure 1: even-numbered steps store and access numerical information, while odd-numbered steps perform the actual operations. This discovery offers an initial blueprint for the internal algorithms of latent reasoning models, contributing to the broader effort of making these opaque yet powerful systems more auditable and reliable.

## 2 Methodology

**Attention-Based Analysis of Latent Steps.** To investigate CODI's use of attention for arithmetic reasoning, we first generated a synthetic dataset based on a "minimal-pair" design. In this design, each pair of math problems shares the same underlying numbers and narrative structure, but a key word or phrase is changed to alter the required arithmetic operation (e.g., "gains 3" vs. "loses 3"), with the constraint that all intermediate and final values in both problem variants remained positive integers. Using this dataset, we executed each problem pair on the model and captured the full attention matrices from every head and layer for each of the six latent reasoning steps ($z_1, \ldots, z_6$) [Shen et al., 2025], for both quantitative and qualitative analysis.

Our quantitative analysis began by measuring the fraction of attention mass that each latent step directs toward numeric tokens within the input prompt. Specifically, for each step, we calculate the ratio of attention on numeric tokens to the total attention on all prompt tokens, and then average this ratio across all layers and attention heads. This metric reveals the temporal dynamics of the model's focus on numerical data. We then identified specialized heads by calculating the proportion of their attention on key numerical operands (e.g., operand $B$ in $A + B$) versus the entire prompt. The minimal-pair design is crucial here, as it isolates how attention to an operand changes when only the operational phrase is altered.

**Sparse autoencoder features.** We train a Sparse Autoencoder (SAE; Bricken et al. 2023) on residual-stream activations from layer 10 of CODI on GSM8K [Cobbe et al., 2021]. The SAE uses a hidden layer $32\times$ the residual width and attains $R^2 = 0.988$ with mean L0$= 7.97\%$. We selected

Layer 10 via a layer-wise $P-P'$ $\Delta$-variance sweep on our minimal-pair runs (variance of $P-P'$ across pairs/latents); it ranked highest (full table in Appendix).

From this dictionary, we constructed conceptual directions by first identifying features corresponding to specific concepts. To do this, we calculated a Cohen's d score for each feature based on its activation for our target concept (e.g., the digit '2') versus others, and selected all features with a score above a pre-defined threshold. The final direction vector was created by summing the decoder weights of these corresponding features. We then probed the model's latent steps $(h_1, \ldots, h_6)$ with these directions both observationally, by measuring cosine similarity, and causally via activation steering [Turner et al., 2023]. For interventions, we added each direction vector to the residual stream—scaled to 5% of the activation's norm at that step—and measured the effect on the output using KL Divergence and the change in log probability ($\Delta$ log-prob).

**Activation Patching.** We use activation patching, a causal localization technique that swaps internal activations with those from a perturbed run, to isolate the causal effects of specific layers and latent thoughts Meng et al. [2022], Heimersheim and Nanda [2024]. The procedure involves a clean run on an original GSM8k question to establish a baseline log probability, followed by a corrupted run on a perturbed version[1] of the question where we cache the residual stream at every layer and latent thought. Finally, in a patching run on the clean input, we replace the residual stream at each position with the corresponding cached stream from the corrupted run. The causal effect is quantified by the difference in the model's average log probability of the correct answer tokens between the clean and corrupted run. This analysis is restricted to questions where the model correctly answers both the original and perturbed versions.

**Early Answer Generation.** Based on a similar experiment from Hao et al., 2024, we probe the model's intermediate reasoning by forcing it to generate an answer early. For a given problem, we let the model generate one latent thought, then insert the end-of-thought token to get its answer. We repeat this process, allowing it to generate two thoughts, then three, and so on, to track how its answer changes with more computation. We also measure approximate problem difficulty via operations required to solve problem (from compressed CoT found in the GSM8k-Aug dataset from Deng et al., 2023)

**Latent Thought Rearrangement.** To evaluate whether latent thoughts are self-contained or sequentially dependent, we systematically manipulate both the order and the number of thoughts available to the model and measure the resulting impact on accuracy. Specifically, we record the model's latent outputs and reintroduce them in controlled configurations, varying one or more of sequence, subset, repetition or composition. Each configuration is tested across the GSM8K dataset and the accuracy of the first-ranked logit is measured.

## 3 Results

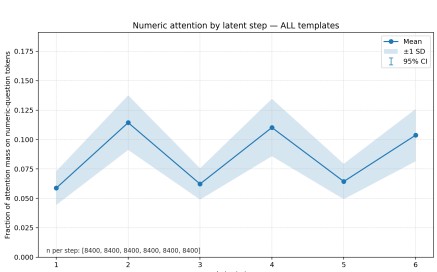

(a) Proportion of attention on all numeric tokens across latent steps.

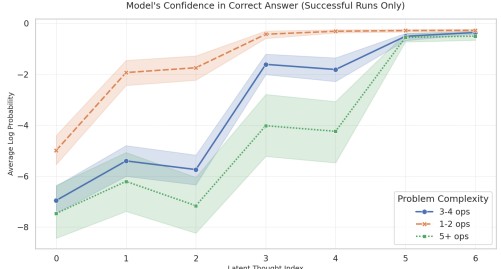

(b) Latent-to-latent average log probability of correct answer in problems the model answered correct, separated by operations required to solve problem

Figure 2: Observational evidence for the alternating roles of latent steps. (a) Attention to numeric tokens peaks on even-numbered steps. (b) The model's attention is consistently directed toward previous even-numbered "scratchpad" steps.

---

[1]Questions are perturbed by changing various numbers in them

Table 1: Number steering: even–odd deltas (even − odd); positive means larger on even steps. Features: Cohen's $d \geq 0.6$; $n = 100$ per digit; uncertainties are $\pm$ error.

| Digit | $|F|$ | $\Delta$COS | $\Delta$KL | $\Delta \log P$ |
|---|---|---|---|---|
| 2 | 237 | $+0.0916 \pm 0.0092$ | $+0.008697 \pm 0.002639$ | $+0.0282 \pm 0.0247$ |
| 5 | 184 | $+0.0824 \pm 0.0109$ | $+0.006080 \pm 0.002762$ | $+0.00227 \pm 0.02596$ |

Table 2: Operation steering: odd–even deltas (odd − even); positive means larger on odd steps. Features: Cohen's $d \geq 0.6$; strength $\gamma = 0.05$; $n = 50$ per operation; $|\cos|$ uses magnitude; uncertainties are 95% CI half-widths.

| Operation | $|F|$ | $\Delta|$COS$|$ | $\Delta$KL | $\Delta \log P$ |
|---|---|---|---|---|
| Subtraction | 1391 | $-0.0334 \pm 0.0093$ | $+0.0180 \pm 0.0099$ | $+0.0726 \pm 0.0531$ |
| Addition | 2315 | $-0.1640 \pm 0.0091$ | $+0.0105 \pm 0.0080$ | $+0.0519 \pm 0.0420$ |

Building on the original CODI paper's observation that some latent thoughts decode to intermediate results while others appear to be "placeholders or transitional states" [Shen et al., 2025], our causal interventions provide strong evidence that CODI follows a structured scratchpad–computation cycle. Even-numbered steps function as scratchpads that store numerical content, whereas odd-numbered steps perform active computation. We first establish the causal localization of these roles using activation patching and SAE-based number steering, then turn to supportive observational patterns.

**Even (scratchpad) steps—numerical content is causally localized.** SAE-based number steering quantifies the asymmetry (Table 1): for both digits tested (2 and 5), even steps show larger shifts in representation alignment ($\Delta$COS) and distributional divergence ($\Delta$KL) than odd steps. The corresponding $\Delta \log P$ effects trend positive but their uncertainties overlap zero, so we treat them as supportive rather than decisive and leave broader digit coverage to future work.

**Odd (computation) steps—operations are causally localized.** Early-answer forcing shows the largest accuracy gains immediately after odd steps (Figure 2b), indicating that key computational updates occur there. SAE-based operation steering quantifies this asymmetry (Table 2)[2]. Odd steps produce larger distributional shifts (KL) than even steps for both subtraction and addition, while projection magnitudes ($|\cos|$) are larger on even steps—consistent with operation structure being loaded on even steps and applied on odd steps.[3] Together with latent-thought rearrangement (Figure 5), these results localize computation to odd steps.

**Observational patterns align with the causal picture.** Without interventions, the model's focus on numeric tokens peaks at even steps (Figure 2a). Latent-to-latent attention preferentially routes to prior even steps (Appendix, Figure 3a), and several attention heads exhibit operand-identification specialization predominantly on even steps (Appendix, Figure 3b). These descriptive signals are consistent with even steps serving as staging points for information that is subsequently used by the odd steps.

## 4 Conclusion and Limitations

This paper provides mechanistic evidence for a structured, algorithmic process within CODI's latent steps. Building on the "placeholder" observation, we show a "scratchpad–computation" cycle: even steps store numeric information; odd steps operate on it. Both observational probes and causal interventions support this alternation, indicating that continuous latent reasoning can be reverse-engineered with mechanistic tools.

Our experiments target one model (GPT-2 CODI) and one domain (GSM8K arithmetic), with SAE probes centered on a single layer and four concepts (the digits "2" and "5" and the operations subtraction and addition) and attention analyses based on limited, templated minimal pairs. The next

---

[2]KL is used as the primary metric for step-local disruption because it captures full-distribution change and yields tighter intervals in our setting; $\Delta \log P$ targets only the correct answer token and is therefore noisier.

[3]Cosine sign depends on direction orientation; we interpret KL and $|\cos|$ magnitude together.

frontier is to test whether this alternating cycle holds in larger models (e.g., LLaMA-style latent reasoning) and beyond arithmetic. This work provides the blueprint for those investigations.

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

# A    Additional Experimental Results

## A.1    Attention Analysis Visualizations

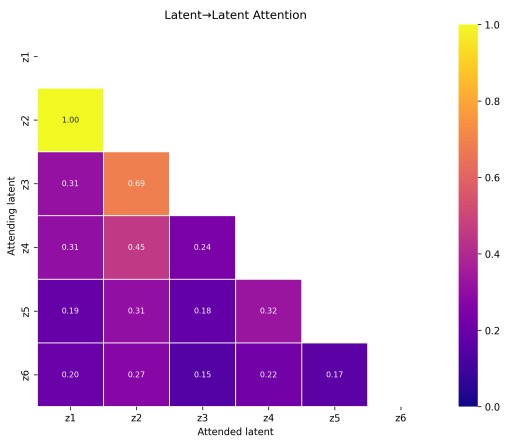

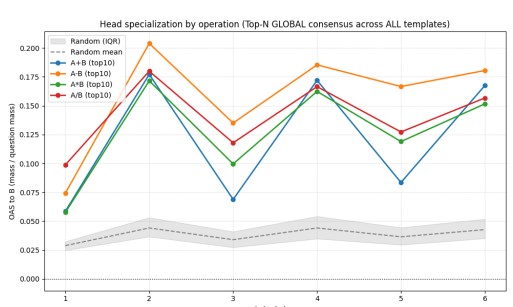

(a) Latent-to-latent attention, averaged across all layers, showing consistent focus on prior even-numbered steps.

(b) Specialization of attention heads for identifying the second numerical operand in arithmetic problems.

Figure 3: Attention analysis results providing further evidence for the scratchpad-computation cycle.

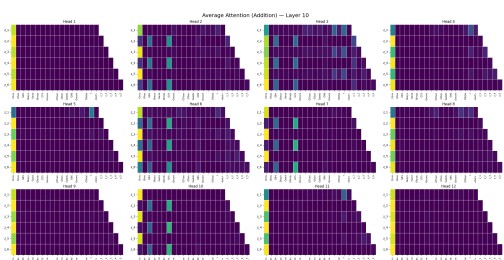

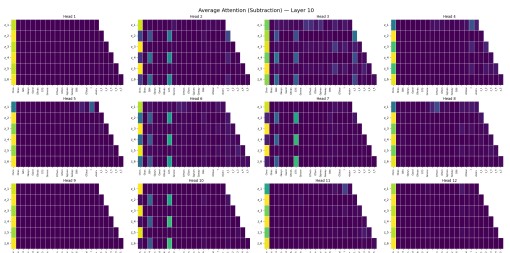

(a) Average attention patterns on addition problems.

(b) Average attention patterns on subtraction problems.

Figure 4: Average attention patterns for 200 examples of the template pair ("[name1] has [A] [item]s and finds [C] more. Then [name2] adds [B] more. Total?", "[name1] has [A] [item]s and finds [C] more. Then [name2] gives away [B]. Total?") across all 12 heads in Layer 8. Note the distinct, sparse patterns in certain heads (e.g., Head 3, Head 7) which focus on key tokens and latent steps.

## A.2    Latent Thought Rearrangement Results

For each question $q$, we first perform a forward pass through the model to extract the sequence of continuous thoughts:

$$Z = (z_1, z_2, \ldots, z_n), \quad z_i \in \mathbb{R}^d.$$

These vectors are recorded and stored as the *original continuous thoughts*. Next, we perform a second forward pass where the same question $q$ is input to the model, but instead of allowing the model to generate its continuous thoughts naturally, we *manually feed* the stored thoughts $Z$ after applying a transformation $\mathcal{F}$. This allows us to evaluate the effect of various thought manipulations on the final output while keeping the model and question fixed.

We experimented with the following set of transformations:

- **Baseline:**

$$\mathcal{F}_{\text{CODI}}(Z) = Z$$

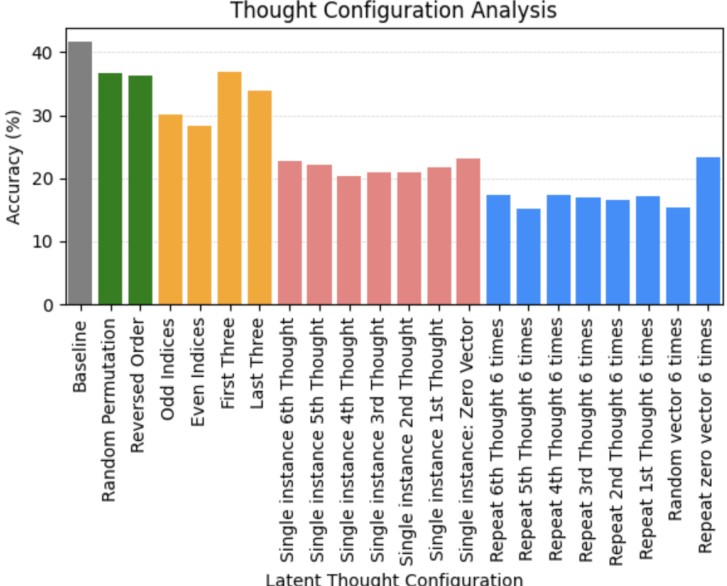

Figure 5: Results for the latent rearrangment experiment.

- **Order manipulations:**

$$\mathcal{F}_{\text{perm}}(Z) = (z_{\pi(1)}, z_{\pi(2)}, \dots, z_{\pi(n)}), \quad \pi \sim \text{Uniform}(\mathcal{S}_n),$$

$$\mathcal{F}_{\text{rev}}(Z) = (z_n, z_{n-1}, \dots, z_1)$$

- **Subset selection:**

$$\mathcal{F}_{\text{odd}}(Z) = (z_1, z_3, z_5), \quad \mathcal{F}_{\text{even}}(Z) = (z_2, z_4, z_6),$$

$$\mathcal{F}_{\text{first3}}(Z) = (z_1, z_2, z_3), \quad \mathcal{F}_{\text{last3}}(Z) = (z_4, z_5, z_6)$$

- **Single-thought substitution:**

$$\mathcal{F}_{k\text{-only}}(Z) = (z_k), \quad k \in \{1, \dots, 6\},$$

$$\mathcal{F}_{\text{zero-only}}(Z) = (\mathbf{0}), \quad \mathbf{0} \in \mathbb{R}^d$$

- **Thought repetition:**

$$\mathcal{F}_{k\text{-repeat}}(Z) = (z_k, z_k, \dots, z_k), \quad |Z| = n,$$

## A.3 Layer-wise $\Delta$-variance Sweep (Paired $P/P'$)

| Layer | Mean $\Delta$-variance |
|---|---|
| 0 | 3.1273 |
| 1 | 2.9035 |
| 2 | 2.8513 |
| 3 | 3.0911 |
| 4 | 3.3083 |
| 5 | 3.6817 |
| 6 | 4.1858 |
| 7 | 5.5977 |
| 8 | 8.4259 |
| 9 | 11.4380 |
| 10 | **19.3736** |
| 11 | 1.9160 |

Table 3: Layer-wise mean $\Delta$-variance (variance of $P-P'$ across pairs and latent steps). Layer 10 is largest.

## A.4 Activation Patching

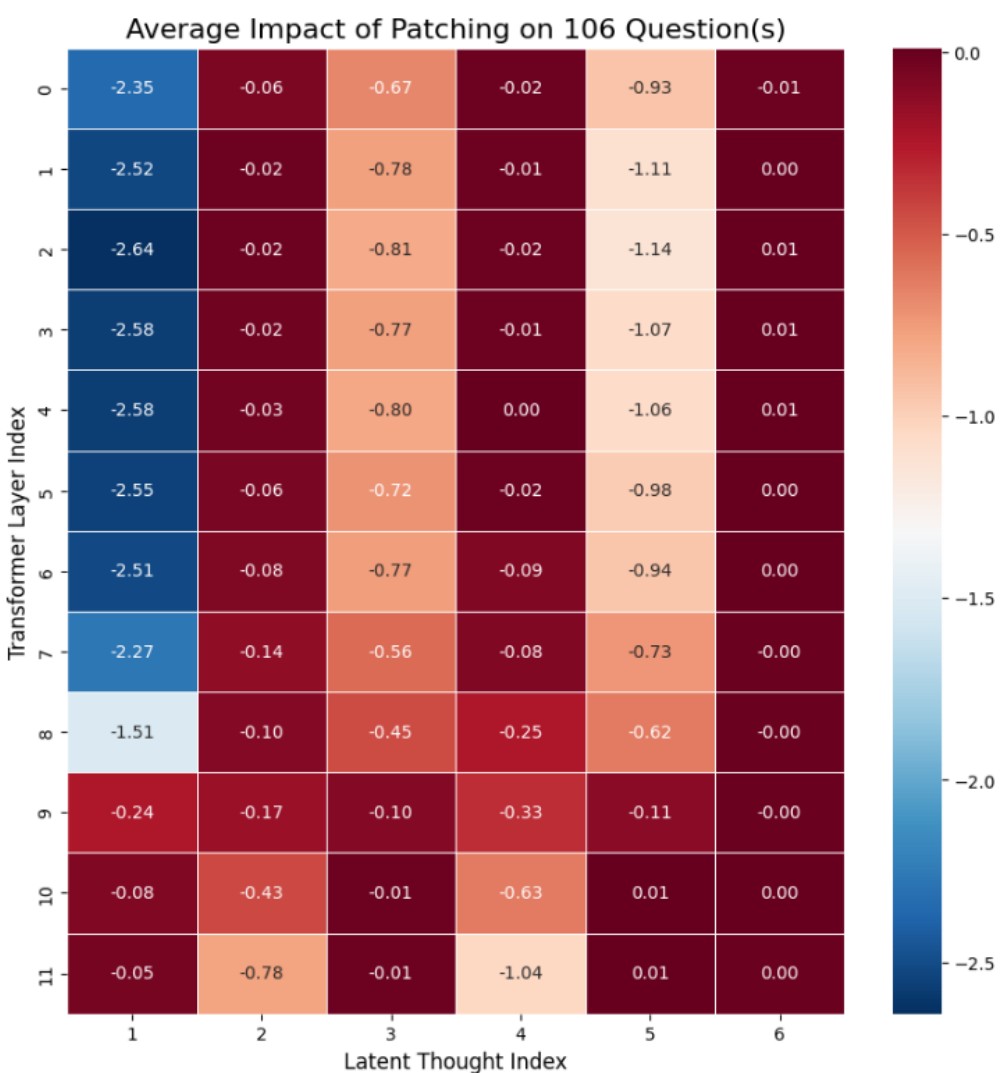

Figure 6: Heatmap comparing the difference of average log probability of the correct numerical answer tokens between the patched and unpatched runs of CODI on questions the model answered correctly.

## A.5 Self-Explanation

To try and interpret the latent thought tokens from CODI, we prompted its original base (GPT-2 or Llama-3.2-1b-Instruct) to generate a natural language explanation for them. We input the sequence of latent thought tokens generated by CODI into the base model, guided by a prompt asking it to generate a natural language explanation of these "abstract ideas."

For both GPT and Llama versions of CODI, the base model failed to produce coherent interpretations, treating the latent tokens as noise. This result may suggest that the fine-tuning process develops a specialized internal syntax for reasoning that is indecipherable to the foundation model.

### A.6 Prompt Modification

### A.6.1 Core Problem Variations

We test the stability of CODI's latent arithmetic reasoning by applying controlled linguistic and structural edits to a small set of seed problems. The goal is to assess whether latent thoughts exhibit a consistent reasoning pattern and how distractors or syntactic changes perturb the internal trace.

**Seeds (from GSM8K).**

1. **Ducks.** Janet's ducks lay 16 eggs per day. She eats three for breakfast every morning and bakes muffins for her friends every day with four. She sells the remainder at the farmers' market daily for $2 per fresh duck egg. How much in dollars does she make every day at the farmers' market?

2. **Robes.** A robe takes 2 bolts of blue fiber and half that much white fiber. How many bolts in total does it take?

3. **Sprints.** James decides to run 3 sprints 3 times a week. He runs 60 meters each sprint. How many total meters does he run a week?

**Variations.** Each seed is edited into ten variants, yielding **33** total prompts (3 originals + 30 variants), covering:

- Explicit vs. implicit numerical framing
- Passive/rephrased grammar
- Added distractor details
- Ratio-/proportion-based reframing

**Protocol.** Prompts are presented one at a time; the model's **seven** latent thoughts (LT0–LT6) are recorded. Accuracy is computed per latent response against ground truth to inspect progression and robustness.

**Results (Exp. 1).**

- **LT0 (Surface bias):** Overconfident, often incorrect guesses driven by salient tokens; distractors amplify errors (e.g., "726", "400").
- **LT1 (Subtraction/filtering):** Encodes reductions/proportions reliably (halves, remainders); robust to modest rewording.
- **LT2 (Hallucination/memorized):** Unstable; implausible values/units appear.
- **LT3 (Final arithmetic logic):** Accurate values in most contexts.
- **LT4–LT6 (Validation/finalization):** Reinforce LT3 and stabilize outputs.

Accuracy rises from ∼20–30% at LT0 to >70% at LT5–LT6 on original/clear prompts, but drops sharply under distractor or proportional edits. Overall, latent thoughts behave like a modular pipeline: LT0 (surface), LT1 (early arithmetic), LT3 (reasoning engine), LT4–LT6 (confirmation). Irrelevance/ambiguity disrupt LT3 and prolong error-prone states (LT0/2).

### A.6.2 Experiment 2: Prompt-Type Comparison

We next examine how prompt categories affect latent-step accuracy.

**Setup.** Ten GSM8K problems (rates, proportions, inventory changes, multi-step sequences) are each modified in four ways (two versions per modification), yielding **90** prompts (10 originals + 80 edits):

1. **Original**: Unmodified GSM8K text
2. **Clear arithmetic**: All operations and numbers stated explicitly
3. **Rephrased/passive**: Syntax altered, semantics preserved
4. **Distractor**: Added plausible but irrelevant details

5. **Proportional/ratio**: Reframed as percentages or ratios

Each prompt is processed individually; LT0–LT6 are recorded; accuracy is computed per latent.

**Results (Exp. 2).**

- **Original**: Reaches ∼90% at LT5–LT6.
- **Clear arithmetic**: Peaks at ∼75%; explicit formatting underperforms natural phrasing.
- **Rephrased/passive**: Early accuracy drops to ∼15%, partially recovers to ∼60% at LT5.
- **Distractor**: 0% at LT0; final accuracy capped at ∼15% (persistent confusion).
- **Proportional/ratio**: ≈0% across all latents.

