# OpenReview forum: "Scratchpad Thinking: Alternation Between Storage and Computation in Latent Reasoning Models"
_NeurIPS.cc/2025/Workshop/UniReps — UniReps2025_

### Official Review · Reviewer_Ddd4 · 2025-09-15

**Confidence:** 3

**Review:**

## Summary
This paper investigates how reasoning is carried out in a specific latent reasoning model, CODI (Continuous Chain-of-Thought via Self-Distillation). Several popular interpretability methods are applied. The main finding is that CODI appears to use a scratchpad-style computation with an alternating cycle: even-numbered steps act as scratchpads for storing numerical information, while odd-numbered steps perform the actual computations.

## Strengths
* The topic is timely. Latent reasoning models offer the promise of more efficient reasoning but lack interpretability, making this a well-motivated research direction.
* The paper is highly relevant to the community, touching on reasoning in latent space and mechanistic interpretability.
* The application of the proposed interpretability methods is described clearly and, as far as I can tell (though I am not an expert in interpretability), is accurate.
* The conclusions drawn from these methods are supported by experimental results.

## Weaknesses
* The paper does not follow the standard ML conference structure, which makes it more difficult to follow. The introduction presents a detailed list of interpretability methods with short explanations of what each does. This would be better placed in the methodology section, while the introduction could provide only a high-level overview. The methodology section also includes not only method descriptions but discussions of how they were applied to obtain results. These belong more naturally in an experimental section, which could be created by merging the methodology and results sections.
* The reliance on a single, relatively small model makes the generality of the findings uncertain. While the conclusions may still hold for larger models and different datasets, it is possible that the alternating behavior between odd and even steps will not generalize.
* The broader impact of the findings is unclear. In particular, the abstract’s claim that “...this work lays the groundwork for auditing latent reasoning models and integrating them more safely into critical applications” feels out of place and insufficiently supported.

**Score:**

3

**Topic Fit:**

2

---

### Official Review · Reviewer_RExZ · 2025-09-15
**Preliminary work on mech interp for latent reasoning, distinct findings**

**Confidence:** 4

**Review:**

**Summary**:

This paper presents a preliminary investigation into the mechanistic interpretability ('mech interp') of a CODI-style latent reasoning model. The original paper hypothesized that there is an alternating scheme between steps, and this paper throws the kitchen sink of mech interp at it to provide evidence that the original hypothesis is correct, and that CODI models do have some distinctive behavior depending on the reasoning step. The paper is short and mainly finds some interesting results, but is limited in the sense that it only examines a single latent reasoning model and only on a limited domain of simple arithmetic tasks.

That being said, latent reasoning models are few and far in between currently, and for an extended abstract track at a workshop this paper is a good fit. I do have some problems with the paper (addressed below), but in my opinion it is a good preliminary work that helps to shore up some upcoming areas of interest in the reasoning and safety communities, namely latent reasoning interp. I recommend acceptance to the workshop.

**Strengths and Weaknesses**:

1. Quality
	1. 3/4. Overall this is a solid treatment with lots of experimental details provided.
2. Clarity
	1. 2/4. I do have some problems with the presentation of the paper, notably how I believed it is not self contained to an adequate degree. See below.
3. Significance
	1. 3/4. There is not much work on mech interp of latent reasoning models, and the authors were able to show some distinct behavior in one model.
4. Originality
	1. 3/4. Similarly to the above, this is a brand new area and so preliminary work here is impactful. Given that this work mainly explores a hypothesis from the CODI paper, and that there is additional (albeit scarce) work on mech interp for reasoning, I don't give a 4/4 here but it is close.

**Questions & Limitations**:
Note that I am combining the questions and limitations section of the NeurIPS reviewer guideline into one, because there is no rebuttal period for the workshop.

Minor concerns:
- On line 73, you indicate that the Cohen's d score must be above a 'pre-defined threshold'. You mention this in the captions for tables 1 and 2, but a mention in this paragraph is a good idea. More importantly, you need to justify your choice here. Choosing hyperparameters is fine (although for a full conference paper some methodological justification through a grid search or something would be good), but at least justify everything here.
- Is there any particular reason to only look at the change in log probability for the correct answer token, instead of the change relative to all of the logits? This is minor, but it could be better justified. For instance, some work on chain of thought faithfulness investigates more granular changes to the logits instead of just answer switches.
- The title mentions 'models', but you only investigate a single model.

Major concerns:
- There is a fair bit of relevant work that is not covered here. Latent reasoning is a very new area, but a few works have started to  come up. Of particular note:
	- The COCONUT paper from Meta, which introduced the more 'canonical' latent reasoning model. See Hao et al., 2024, ''Training Large Language Models to Reason in a Continuous Latent Space". https://arxiv.org/abs/2412.06769
	- A recent position paper on chain of thought monitorability, particularly the potential limitations posed by latent reasoning models for monitoring and why interp on them is direly needed as they become more prominent. See Korbak et al., 2025, "Chain of Thought Monitorability: A New and Fragile Opportunity for AI Safety". https://arxiv.org/abs/2507.11473
	- The most relevant paper to your own is something that has done mech interp on LLM reasoning, although not necessarily on latent reasoning models. See Bogdan et al., 2025, "Thought Anchors: Which LLM Reasoning Steps Matter?". https://arxiv.org/abs/2506.19143
- The paper is not as self-contained as it should be. I understand that it's a four page workshop paper and that it introduces a lot of new experiments, but I was not familiar with CODI and in order to better understand your paper, I had to look this up and read through it to understand how the model worked. I'm most familiar with COCONUT-style latent reasoning models, where a vector is passed through each step instead of a token, so the student-teacher paradigm of CODI and how it reasons in latent space is new to me. At least this can be an additional appendix, with perhaps 2-3 paragraphs and one figure. But to make your paper more clear to those interested, a very brief overview is a good idea.

So overall I think it is a fine preliminary work, and it's exploring an area where there has been hardly any work so far. It is definitely limited in scope, and there are a few issues which I've just noted, but this is a good fit for the workshop and I'm happy to recommend its acceptance. Moreover, I think that better understanding latent reasoning models is of obvious importance to society and safety and I am glad to see research here. Good work.

**Overall Score**:

4/5. Accept.

**Confidence**:

4/5

**Score:**

4

**Topic Fit:**

2

---

### Official Review · Reviewer_DgYZ · 2025-09-16
**Promising idea on latent reasoning, but unclear presentation and limited insight**

**Confidence:** 4

**Review:**

This submission examines latent reasoning models, focusing on CODI, through mechanistic interpretability tools like sparse autoencoders, activation patching, attention analysis, and causal interventions. The key claim is that CODI alternates between "scratchpad steps" for storing intermediate values and "computation steps" for arithmetic operations. If validated, this finding could enhance our understanding of these opaque models and improve safety.

However, the abstract is challenging to follow. Many sections are vague and lack crucial details. It often reads like a list of methods without sufficient explanation for their selection, application, or insights gained. The experimental results are scattered and difficult to interpret, raising questions about the strength of the evidence for the claimed alternation.

**Strengths:**
- Addresses an important problem: improving the interpretability of latent reasoning models.
- Applies a toolkit of interpretability methods to a novel case study.
- The hypothesis of alternation is intriguing and could inspire future research.

**Weaknesses:**
- The writing lacks clarity and coherence.
- Many claims lack supporting evidence or quantification.
- The experimental setup is vaguely described, limiting the assessment of controls.
- Relies heavily on existing interpretability tools, showing limited novelty.
- Results focus solely on one model (GPT-2 CODI) and a narrow range of tasks.

**Score:**

2

**Topic Fit:**

2